# Neoadjuvant Immunotherapy for Localized Pancreatic Cancer: Challenges and Early Results

**DOI:** 10.3390/cancers15153967

**Published:** 2023-08-04

**Authors:** Robert Connor Chick, Andrew J. Gunderson, Shafia Rahman, Jordan M. Cloyd

**Affiliations:** 1Department of Surgery, Division of Surgical Oncology, The Ohio State University Wexner Medical Center, Columbus, OH 43210, USA; 2Department of Medicine, Division of Medical Oncology, The Ohio State University Wexner Medical Center, Columbus, OH 43210, USA

**Keywords:** pancreatic cancer, immunotherapy, neoadjuvant, immune checkpoint inhibitor, cancer vaccine, clinical trial, review

## Abstract

**Simple Summary:**

Pancreatic cancer is a deadly disease with limited effective treatments. Despite growing interest in immunotherapy for other cancer types, immunotherapy has not shown significant utility in metastatic pancreatic cancer. However, investigators are examining the role of immunotherapy in the preoperative setting in order to prime the immune system to detect and prevent micrometastatic disease and recurrence. This scoping review describes 9 published trials of preoperative immunotherapy and summarizes 27 ongoing trials using preoperative immunotherapy, generally in combination with other standard neoadjuvant therapy. The results of these early trials have been promising but have not conclusively established a role for preoperative immunotherapy in pancreatic cancer. The results of the ongoing trials may be able to demonstrate whether combination immunotherapy prior to surgery can improve long-term outcomes.

**Abstract:**

Pancreatic ductal adenocarcinoma (PDAC) is a highly lethal disease due to its late presentation and tendency to recur early even after optimal surgical resection. Currently, there are limited options for effective systemic therapy. In addition, PDAC typically generates an immune-suppressive tumor microenvironment; trials of immunotherapy in metastatic PDAC have yielded disappointing results. There is considerable interest in using immunotherapy approaches in the neoadjuvant setting in order to prime the immune system to detect and prevent micrometastatic disease and recurrence. A scoping review was conducted to identify published and ongoing trials utilizing preoperative immunotherapy. In total, 9 published trials and 27 ongoing trials were identified. The published trials included neoadjuvant immune checkpoint inhibitors, cancer vaccines, and other immune-modulating agents that target mechanisms distinct from that of immune checkpoint inhibition. Most of these are early phase trials which suggest improvements in disease-free and overall survival when combined with standard neoadjuvant therapy. Ongoing trials are exploring various combinations of these agents with each other and with chemotherapy and/or radiation. Rational combination immunotherapy in addition to standard neoadjuvant therapy has the potential to improve outcomes in PDAC, but further clinical trials are needed, particularly those which utilize an adaptive trial design.

## 1. Introduction

Pancreatic ductal adenocarcinoma (PDAC) remains among the leading causes of cancer-related death in the United States and around the world [1]. For the relatively small proportion of patients who present with localized disease, many are found to have occult metastatic disease and others recur within a short time after surgery [2]. For these reasons, novel systemic therapies are urgently needed to improve the outcomes of patients with PDAC. Progress in developing effective therapies for advanced PDAC has been slow, in part due to the high lethality of advanced PDAC, though this limitation may become less important as molecular and genomic technologies are able to increase the rate of development of novel therapies. Despite significant advances in the application of immune oncology for other cancer types, immunotherapy has not been firmly established as a treatment modality in PDAC [3,4]. While there are several reasons for this, the leading hypothesis is that PDAC tends to be an immunologically “cold” tumor, characterized by poor lymphocyte infiltration in the tumor microenvironment (TME), with poor response to current immunotherapy approaches [5,6].

PDAC has been described as having an immune privileged TME [7]. In addition to being lymphocyte poor, the tumor stroma is characterized by prominent M2 macrophages, which suppress rather than activate the acquired immune system, and regulatory T-cells (T_reg_), which limit the development of cell-mediated immunity [8]. This is in part due to the lower tumor mutational burden (TMB) in PDAC, which has been suggested to be a driving factor in immunogenicity and susceptibility to immune checkpoint inhibition (ICI) [6]. However, preclinical data as well as clinical trial data in the metastatic setting suggest that cytotoxic chemotherapy may increase the immunogenicity of PDAC by converting the TME from immunologically “cold” to “hot” [9,10]. Other preclinical data suggest that combination therapy directed against multiple immune-related targets may be the key to realizing the benefit of immunotherapy in PDAC, such as targeting myeloid-derived suppressor cells (MDSC) in addition to immune checkpoints, since these cells are thought to play a key role in the maintenance of lymphocyte-poor tumor stroma [11]. Additionally, emerging evidence highlights subsets of PDAC that, rather than being lymphocyte poor, demonstrate lymphocytic infiltration and tertiary lymphoid structures; these are correlated with longer survival, supporting the hypothesis that the immunologically “cold” TME is a major factor contributing to poor outcomes in this disease [12,13]. Nevertheless, poor objective response rates have been observed in clinical trials of immunotherapy for advanced PDAC [14,15]. More broadly, various iterations of traditional chemotherapy with or without radiotherapy or targeted therapies have been unable to meaningfully improve outcomes. Therefore, the addition of immunotherapy may provide a much-needed breakthrough in long-term outcomes in localized PDAC.

While surgical resection is generally considered necessary for curative-intent treatment, systemic therapies are increasingly administered prior to surgery for patients with localized PDAC [16,17]. Reasons for recommending neoadjuvant therapy (NT) include the early treatment of presumed micrometastatic disease, improved rates of downstaging and margin-negative resection, and ensuring delivery of effective therapy given the challenges of administering adjuvant chemotherapy after major pancreatic surgery [18]. Systemic therapies first established in the metastatic setting (i.e., 5-fluorouracil, leucovorin, irinotecan, and oxaliplatin (FOLFIRINOX) or gemcitabine and nab-paclitaxel (GA)) have been adopted as first-line options in the neoadjuvant setting [19]. Thus, given the intense interest in developing immunotherapy approaches for patients with metastatic PDAC, investigators are exploring their utility in the neoadjuvant setting as well. Theoretical advantages of delivering immunotherapy prior to surgery include improved immune surveillance through increased recognition of tumor antigens, mitigation of perioperative immunosuppression, and the ability to test histologic responses via post-surgical pathology [20,21,22,23].

Despite the importance of this topic and the accumulation of emerging data, the existing literature has not been previously systematically summarized. Therefore, the purpose of this review is to summarize the results of early clinical trials of neoadjuvant immunotherapy in PDAC and to describe the current landscape of ongoing registered prospective trials.

## 2. Materials and Methods

This scoping review was conducted according to existing guidelines [24]. Using the search terms neoadjuvant, preoperative, perioperative, pancreatic cancer, pancreatic ductal adenocarcinoma, and immunotherapy, MEDLINE and clinicaltrials.gov were searched for published and ongoing clinical trials, respectively, of studies reporting the outcomes of patients receiving immunotherapy prior to planned pancreatectomy for localized PDAC. The search strategy was adopted from a recent review of perioperative immunotherapy in gastroesophageal junction cancers [21]. Abstracts were manually screened for inclusion and exclusion criteria. Inclusion criteria consisted of English language full-length articles published after 2010 that reported outcome data, whereas non-English language papers, metastatic settings, published protocols, research letters, unpublished abstracts, and preclinical studies were excluded. Trials where immunotherapy was also given postoperatively were included, as long as immunotherapy was initiated prior to surgery. Reference lists were screened to identify additional relevant articles. Clinical trial registration numbers were cross-referenced with published trials to avoid any duplicates. Titles and abstracts were screened for exclusion criteria followed by a full-text review of articles that met the inclusion criteria. A descriptive analysis of each included study was performed, including methods, participants, interventions, and outcomes. Due to the paucity and heterogeneity of published trials of neoadjuvant immunotherapy in pancreatic cancer, a systematic review and meta-analysis were not performed.

## 3. Results

Of the initial MEDLINE search results (*n* = 30), 10 publications of nine trials (one trial published both early and long-term data) met all inclusion criteria. Study types included one phase III randomized controlled trial, one nonrandomized single-arm phase II trial, one phase I/IIa trial, three phase I trials, and a relevant case report. In addition, from the initial clinicaltrials.gov search results (*n* = 250), 27 registered trials met the inclusion criteria and were included in this review. Trial types consisted of prospective clinical trials, ranging from phase I to phase III, of checkpoint inhibitors alone or in combination with other immune therapies; phase I and II trials of vaccines; and phase I studies of agents directed at other immune-related targets. Published studies were categorized according to type of immunotherapy (immune checkpoint inhibition, vaccines, and other immune-modulating biologic therapies) and further discussed below. The included studies are summarized in Table 1.

### 3.1. Immune Checkpoint Inhibition

Only three publications reported the use of neoadjuvant immune checkpoint inhibition (ICI): one case report of neoadjuvant ICI combined with stereotactic body radiotherapy (SBRT), a phase II prospective trial of neoadjuvant tislelizumab plus GA with SBRT, and a third recently published trial combining a cancer vaccine with ICI and another immune-stimulating monoclonal antibody.

The case report described an 83-year-old woman with a poor performance status and locally advanced PDAC whose tumor was microsatellite unstable (MSI-H) and stained positively for programmed death receptor ligand 1 (PD-L1). She was treated with SBRT followed by pembrolizumab (a programmed cell death protein 1 (PD-1) inhibitor). After a partial radiographic response, she underwent distal pancreatectomy, splenectomy, en-bloc wedge gastrectomy and segmental colectomy and was found to have had a pathologic complete response [25].

Du et al. recently published an exploratory single-arm phase II trial of perioperative tislelizumab (a PD-1 inhibitor) in combination with GA and SBRT. Prespecified exploratory analyses included eosinophil count (which is associated with response to immunotherapy in triple-negative breast cancer) [33], CA 19-9, circulating tumor DNA (ctDNA), the neutrophil-to-lymphocyte ratio (NLR), and TMB. Of 29 patients with locally advanced (LA) or borderline resectable (BR) PDAC, 25 completed therapy. The objective response rate (ORR) was 60% and the disease control rate (DCR) was 100%. Overall survival (OS) at 12 months was 72% and progression-free survival (PFS) was 64%. No serious immune-related adverse events (irAE) were reported. Exploratory analyses demonstrated improved PFS for those with elevated eosinophil counts, a decline in CA 19-9, and a decline in ctDNA. There were no survival differences between those with high or low NLR ratios or TMB [26].

A third study evaluated GVAX, an allogeneic pancreatic cancer vaccine in combination with either nivolumab (PD-1 inhibitor) or both nivolumab and urelumab (an agonist of the T-cell costimulatory molecule CD137). Subjects received a single cycle of immune therapy prior to surgery, then proceeded with adjuvant treatment which included their assigned immune therapy plus standard-of-care chemotherapy. The combination of GVAX, nivolumab, and urelumab met the primary endpoint of generating CD137^+^ CD8^+^ T cells. Though underpowered to detect a statistically significant survival difference, there was an impressive numerical improvement in median disease-free survival (DFS) from 15.0 months for GVAX + nivolumab (*n* = 18) to 33.5 months for the three-drug combination (*n* = 11). Similarly, the median OS was 27.0 months for GVAX + nivolumab vs. 35.6 months for the three-drug combination. Perhaps most interesting is the neoadjuvant platform trial design which allows for simultaneous comparison of different combinations or sequences of immune therapies [27].

There are 17 planned or ongoing trials of neoadjuvant ICI, mostly combined with standard of care chemotherapy and/or radiation (*n* = 11) or in combination with other immunomodulating agents (*n* = 6). The CIPSD-4 trial, estimated to be completed in April 2024, is a phase III trial of a perioperative PD-1 inhibitor in combination with FOLFIRINOX (NCT03983057) [34]. There are five phase II studies, summarized in Table 2, using durvalumab, a PD-L1 inhibitor; pembrolizumab; and sintilimab, another PD-1 inhibitor. One phase II study was terminated for safety (NCT03344172). There are five earlier phase studies (phase I/II) summarized in Table 2.

Studies of ICI in combination with other immunotherapies are generally newer. One phase II trial (NCT03161379) of neoadjuvant nivolumab with GVAX (GM-CSF secreting allogeneic pancreatic tumor cell vaccine) and cyclophosphamide has completed accrual and is in the follow-up period. Another phase II trial (NCT03563248) combined neoadjuvant nivolumab with losartan and SBRT in BR/LA PDAC; preliminary results have been presented but are pending publication [35]. Three other phase II trials are currently recruiting and are summarized in Table 3. Two phase I trials have been completed and are awaiting publication: one combined neoadjuvant pembrolizumab, IMC-CS4 (CSF-1R inhibitor), GVAX, and cyclophosphamide (NCT03153410) and the other combined neoadjuvant nivolumab, ipilimumab (cytotoxic T-lymphocyte associated protein 4 (CTLA4) inhibitor), and pepinemab (SEMA4D inhibitor) (NCT03373188) [36].

### 3.2. Vaccines

In addition to the platform trial of GVAX, nivolumab, and urelumab described above, there are two published studies of cancer vaccine therapy in the preoperative setting: one using Hyper-Acute-Pancreas algenpantucel-L (HAPa) and the other using GVAX. The stated aim of these therapies is to increase the immunogenicity of in situ PDAC prior to surgical resection.

HAPa is a cancer vaccine consisting of allogeneic pancreatic cancer cells engineered to express the murine alpha-(1,3)-galactosyltransferase gene. Hewitt et al. published a multicenter, phase 3, open label, randomized trial comparing neoadjuvant chemotherapy plus chemoradiation with or without HAPa in 303 patients with BR/LA unresectable PDAC. No significant differences were seen in the primary endpoint of OS (14.3 months for HAPa vs. 14.9 months for control) or in the secondary endpoint of PFS (12.4 months vs. 13.4 months). Only 25% of participants became eligible for surgical resection, with no between-group difference in conversion to resectability [28].

GVAX is an allogeneic GM-CSF-secreting pancreatic cancer vaccine that was studied in a window-of-opportunity trial where participants were randomized to GVAX alone, GVAX with cyclophosphamide at a standard immunomodulatory dose, and GVAX with 2 week cycles of cyclophosphamide. In 2014, Lutz et al. reported changes in the TME after vaccination. This initial study found that 85% of vaccinated, resected tumors had intratumoral tertiary lymphoid aggregates (TLA), a finding thought to be attributable to GVAX and associated with response to immunotherapy in other cancers [37] and survival in PDAC [13,38]. This correlative data demonstrated that T-cell trafficking into the TME to form these lymphoid aggregates is associated with evidence of early T-cell activation, recruitment of T_reg_, and upregulation of interferon-gamma and the PD-1/PD-L1 pathway [39]. In 2021, Zheng et al. published the long-term results of the trial. Of the 87 participants who were randomized, 76 underwent pancreatectomy. Those who received GVAX alone had a DFS of 18.4 months, which was significantly longer than the DFS for those who received GVAX with intermittent cyclophosphamide (7.2 months) (*p* = 0.02). There was no significant difference in long-term survival when stratified by density of TLA. A post hoc analysis comparing the outliers in terms of survival showed that those who survived less than 15 months had a much lower TLA density than those who survived more than 24 months [29].

There are two ongoing studies of preoperative vaccine/cell-based therapy in resectable PDAC. A completed phase I study of AdV-tk (herpes viral vector) and valacyclovir in combination with chemotherapy and radiation is currently enrolling participants (NCT02446093). Another study is examining the adoptive transfer of tumor antigen specific T-cells (NCT03192462).

### 3.3. Other Immune-Modulating Biologic Therapies

Although ICI is the most broadly studied mechanism of immunotherapy, other targets are emerging in PDAC [14]. Inhibition of chemokines such as CCR2 and CCR5 has been proposed as a way to reduce the prevalence of T_reg_ and MDSC in the TME [40]. In a phase Ib study of FOLFIRINOX with or without PF-04136309 (a CCR2 inhibitor), 39 patients with BR/LA PDAC were treated with FOLFIRINOX and PF-04136309. At a median follow-up of 72 days, the ORR was 49% with a DCR of 97% [30]. The combination of CCR2 and CCR5 inhibition is being actively studied in later phase trials as described below.

Targeted immunotherapy against cancer antigen 125 (CA-125), a glycoprotein antigen expressed by some pancreatic ductal adenocarcinomas, was studied in a phase I trial of the anti-CA-125 monoclonal antibody oregovomab. Its stated mechanism is the formation of immunogenic antibody–antigen complexes that can be presented by dendritic cells to generate an anti-CA-125 T cell response. Participants were treated with three cycles of 5-FU, leucovorin, gemcitabine, and oregovomab followed by SBRT (40 Gy in five fractions) preoperatively. Of the nine who received chemo-immunotherapy, three proceeded to surgery. Those who received all planned doses of oregovomab had a median OS of 21 months. Two of five patients who received oregovomab developed anti-CA-125 CD8^+^ T-cells. The trial was closed prematurely due to the change in standard-of-care chemotherapy to either FOLFIRINOX or GA [31].

Another immune-related target that has been tested in the neoadjuvant setting is CD40, which is a TNF-like receptor that, similar to CD137, contributes to T cell co-stimulation. Agonism of CD40 has been shown to activate cross-presenting dendritic cells, re-educate tumor macrophages toward an M1 phenotype, decrease collagen deposition in the tumor stroma, and trigger antitumor T cell responses [32]. In an open-label phase I clinical trial at four sites in the US, neoadjuvant GA was given with or without the CD40 agonist selicrelumab to 16 participants with resectable PDAC. For the 15 who underwent pancreatectomy, OS was 23.4 months and DFS was 13.8 months. Examination of the TME on surgical pathology showed that 82% of selicrelumab-treated tumors were T-cell enriched, compared to 37% of untreated tumors (*p* = 0.004) and 23% of chemotherapy-treated tumors (*p* = 0.012). Additionally, fewer M2 macrophages were seen in selicrelumab-treated tumors, fibrosis was reduced, and intratumoral DCs were more mature.

Another trial of neoadjuvant selicrelumab with chemotherapy and radiation recently completed enrollment and is in the follow-up period (NCT01456585). An additional CD40 agonist, CDX-1140, is being studied in combination with CDX-301, a recombinant form of the cytokine Flt3L, notably without additional neoadjuvant chemotherapy (NCT04536077). Finally, NIS793, a TGF-B antagonist, will be studied in combination with neoadjuvant chemotherapy and radiation, but this trial has not yet enrolled (NCT05546411).

## 4. Discussion

Despite its recent successes in other cancer types, the progress in applying immunotherapy approaches for advanced PDAC has been limited. Unlike immunogenic cancers such as melanoma and small cell lung cancer, advanced PDAC has not been shown to respond well to ICI monotherapy in preclinical or clinical trials [4,14]. Indeed, even among MSI-H PDAC (which comprises 1% of all PDAC), the efficacy of ICI has been limited [41,42]. There may be a role for combining chemotherapy and ICI in advanced PDAC as seen in the recent phase 2 PRINCE trial, which also examined the role of the CD40 agonist sotigalimab when added to ICI and chemotherapy [43]. The COMBAT trial of combined PD-1 and CXCR4 inhibition also demonstrated potential benefit in the metastatic setting [44]. Another intriguing combination is that of ICI with epigenetic modulation, with targets such as DNA hypomethylation and histone deacetylase, though this has only been studied in the metastatic setting with limited success [45,46]. Further studies are certainly needed, likely using adaptive or platform trial design, to answer additional questions about treatment sequencing and rational combinations of immune-modulating agents with or without cytotoxic chemotherapy agents.

Despite this limited progress, there is still considerable interest in bringing immunotherapy to the neoadjuvant setting for patients with PDAC. The goals of NT differ based on the anatomic stage of the primary tumor. For BR/LA tumors, neoadjuvant approaches are necessary to generate downstaging that results in successful margin-negative resection. On the other hand, the aim of NT for potentially resectable (PR) tumors is the early treatment of micrometastatic disease. Ensuring completion of perioperative chemotherapy and assessing the in vivo efficacy of chemotherapy are applicable to all anatomic stages of PDAC. Prospective randomized trials have demonstrated that patients are more likely to receive systemic therapy when given preoperatively, without compromising surgical outcomes [47]. At the same time, severe complications during NT can occur; efforts to improve outcomes, decrease attrition, and ensure optimal treatment sequencing are needed [48].

With further understanding of the underlying differences between immunogenic and non-immunogenic PDAC tumors, there is hope and promise that immunotherapy will aid in the realization of all of these goals when combined with systemic and regional therapies in the neoadjuvant setting. In addition to these goals of NT more broadly, preoperative immunotherapy is intended to prime the immune system to detect and eliminate recurrence or metastasis. Delivering immunotherapy in the neoadjuvant rather than adjuvant setting may achieve a more comprehensive antitumor immune response by exposing the adaptive immune system to more tumor antigens. This can be measured by examining the TME in the pathologic specimen. There is also growing evidence for the hypothesis that leaving lymph nodes in situ during neoadjuvant immunotherapy may further enable immune priming, whereas giving these agents after surgery or even after regional radiation may miss this opportunity [49].

Immune surveillance of metastatic disease remains a complex topic that is not fully understood. Recently, however, Cañellas-Socias et al. described a specific immune-related mechanism for latent recurrence in colorectal cancer. Using single-cell transcriptomics in a mouse model, a specific tumor cell type associated with metastatic recurrence was identified. Micrometastases with this cell type tended to be T-cell rich but became T-cell depleted with further growth. Furthermore, these cells were eliminated with neoadjuvant immunotherapy, suggesting that the immature TME associated with metastatic deposition in foreign organs made them susceptible to ICI [50]. This important preclinical work represents a significant step towards understanding the interaction between immune surveillance and metastasis, though more such work is needed in PDAC.

Cancer vaccines have been studied in various disease sites in the metastatic setting with largely disappointing results, but some have shown promise in the adjuvant setting [51]. One proposed explanation for the number of negative trials of vaccine monotherapy in advanced cancer is that vaccines take time to elicit an immune response and may work by changing the growth kinetics of an already slow-growing tumor, whereas in diseases like advanced PDAC, this effect is not enough to overcome rapid or widespread progression [52]. Another problematic phenomenon is that of pseudoprogression, which describes the appearance of progression by typical RECIST criteria as a result of increased immune infiltration of an in situ tumor [53,54]. This could become an issue in the neoadjuvant setting when preoperative restaging is performed. The major advantage, however, of a neoadjuvant vaccination approach is the idea of priming the immune system to recognize tumor-associated antigens and promote immune surveillance and elimination of future metastatic recurrence [51,55]. This aligns well with the idea of “making a cold tumor hot”, which may be necessary but not sufficient to induce an immune response to PDAC [56]. Additionally, the window-of-opportunity design used in the GVAX trials may be a very useful way to evaluate the changes in the TME [57]. These insights could help not only to generate hypotheses, but also to guide personalized adjuvant therapies based on the response to neoadjuvant immunotherapy. A recently published phase I study of cevumeran, an adjuvant, personalized mRNA neoantigen vaccine, in combination with atezolizumab (anti-PD-L1) demonstrated immunogenicity and safety; if biopsy specimens would be adequate for vaccine production, neoadjuvant use could be considered [58].

Various other immune targets are being explored, particularly in combination with each other [56,59]. If neoadjuvant immunotherapy in PDAC has a potential role in future treatment algorithms, it will likely require combination immunotherapy to make a “cold” TME “hot” and then prompt the immune system to survey for recurrence. Of particular interest are combinations of ICI and vaccines. The concept of “pressing the gas” and “releasing the brake” has been demonstrated in preclinical data and there have been suggestions of its efficacy within other trials, but this approach has not yet been well studied in PDAC [60,61]. Results of combination studies using ICI and vaccines will be eagerly anticipated [60,62]. Other combinations use a variety of mechanisms to manipulate the TME, such as combining PD-1 with PD-L1 antagonism to increase effector T cells [63], ICI with SEMA4D inhibition to enable leukocyte penetration into the tumor and reduce M2 tumor-associated macrophages and MDSC [36,64], ICI with CCR2/CCR5 inhibition to enable chemotaxis [30], and ICI with manipulation of the gut microbiome [65]. Interestingly, IL-2 is being re-examined as a component of combination immunotherapy, though initial results in human trials have not been promising [66]. Finally, a clinical and immune response was achieved in a metastatic PDAC patient receiving adoptive transfer of autologous T cells genetically engineered for reactivity against KRAS-G12D [67]. Although these earlier stage results warrant further investigation, these successes indicate that T cell immunity can be therapeutically harnessed in PDAC in certain contexts.

This review has several limitations. The trials included are heterogeneous, small, and hypothesis generating but inconclusive. This is not a systematic review; no formal evaluation of the quality of evidence was performed and some relevant studies may have been inadvertently excluded. Finally, this is a rapidly evolving field with some intriguing but immature data that may become available before publication of this review.

## 5. Conclusions

While few trials of neoadjuvant immunotherapy have been conducted for localized PDAC, the current review suggests there remains strong interest in this treatment paradigm given the number of ongoing clinical trials. While future breakthroughs in immunotherapy for this lethal disease will likely occur in the metastatic setting, investigators will likely aim to bring these to the neoadjuvant setting in order to deliver these promising agents, even to patients with localized cancers, as early as possible.

## Figures and Tables

**Table 1 cancers-15-03967-t001:** Summary of included studies.

Author	Clinical Trial Registration	Immune Agent(s)	Number of Subjects	Primary Outcome
McCarthy et al., 2021 [25]	n/a (case report)	Pembrolizumab	1	n/a
Du et al., 2023 [26]	ChiCTR2000032955	Tislelizumab	29	Not stated
Heumann et al., 2023 [27]	NCT02451982	GVAX, nivolumab, urelumab	46	Intratumoral CD8^+^ CD137^+^ T cells (met)
Hewitt et al., 2022 [28]	NCT01836432	Algenpantucel-L	303	OS (did not meet)
Zheng et al. 2021 [29]	NCT00727441	GVAX	87	Mesothelin specific T-cell response (unpublished)
Nywening et al., 2016 [30]	NCT01413022	PF-04136309	47	Dose/Safety
Lin et al., 2019 [31]	NCT01959672	Oregovomab	11	Disease progression (stopped early)
Byrne et al., 2021 [32]	NCT02588443	Selicrelumab	16	Safety

**Table 2 cancers-15-03967-t002:** Summary of planned or ongoing registered trials of an immune checkpoint inhibitor alone or in combination with chemotherapy and/or radiation.

NCT Number	Phase	ICI	Other Modalities	Status
NCT03572400	II	Durvalumab	Chemotherapy	Recruiting
NCT05132504	II	Pembrolizumab	Chemotherapy	Recruiting
NCT05462496	II	Pembrolizumab	Antibiotics	Not yet recruiting
NCT05562297	II	Sintilimab	None	Not yet recruiting
NCT03563248	II	Nivolumab	Chemotherapy, Radiation, Losartan	Active, not recruiting
NCT03245541	I/II	Durvalumab	Radiation therapy	Not yet enrolling
NCT02305186	I/II	Pembrolizumab	Chemotherapy and radiation	Enrolling
NCT04247165	I/II	Nivolumab, ipilimumab	Chemotherapy and radiation	Enrolling
NCT02930902	I/II	Pembrolizumab	Chemotherapy	Not yet enrolling
NCT03970252	I/II	Nivolumab	Chemotherapy	Enrolling

**Table 3 cancers-15-03967-t003:** Summary of planned or ongoing registered trials of immune checkpoint inhibitor in combination with other immunomodulating agents.

NCT Number	ICI	Other Immunotherapy
NCT04940286	Durvalumab	Oleclumab (CD-73 inhibitor)
NCT03727880	Pembrolizumab	Defactinib (FAK inhibitor)
NCT03767582	Nivolumab	BMS-813160 (CCR2/CCR5 inhibitor), GVAX

## Data Availability

Data are contained within the article.

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
