# Peer review of "Neoadjuvant Immunotherapy for Localized Pancreatic Cancer: Challenges and Early Results"

_cancers, 2023, doi:10.3390/cancers15153967_

Round 1
Reviewer 1 Report
Dear Authors,
First of all, congratulations for your interesting work. I hope that my hints will help you in the next steps of improvement and the final manuscript will be really valuable for the readers. Preoperative immunotherapy is a wonderful topic and I totally agree it should be developed. General comment: too little anout the reasons of slow progress in PDAC therapy advancements; for example, one of the reasons why the progres in case of the PDAC is so slow is the estimated lifespan of patients: for example, before we can perform WGS, most of them die... But recent advancements in WGS speed and cost lowering, namely the nanopore long-reads technology, offers a brand new oportunities and it looks like the PDAC landscape may be changed. I think this issue should be discussed and I strongly encourage to add it to your good paper.
--> a list of abbreviations used would be appreciated
Lines 52-54 --> not everyone may understand why these cells are "not good" for the tumour and its therapy; I suggest a sentence of explanation
Lines 61-62 --> why targeting myeloid cells might be a good option? what they do?
Reviewer 2 Report
I would like to express my gratitude for the opportunity to review the paper titled "Early Results and Ongoing Trials of Preoperative Immunotherapy in Pancreatic Ductal Adenocarcinoma" by Dr. Robert Connor Chick. The paper presents valuable insights into the early results of preoperative immunotherapy trials and outlines the ongoing research in this area.
This review is well-written and pointed out important topics on this matter. I have several concerns.
Minor points
Abstract
# Page1, Line30: There seems unnecessary “,” here.
Methods
# Page 3, Lines 1-3: I appreciate the authors' efforts in providing a clear methodology for this review. However, to improve the comprehensibility of the paper, it would be beneficial if the authors could include the specific search strategy they employed during the literature review.
#Results
# Page4, Line164: There seem to be “five” phase II studies in Table 1 to me.
#Discussion
# Although the efficacy is a big concern, surgical outcomes after neoadjuvant treatment is also an area of investigation. I understand that this is not a topic for this review, but I would like to suggest the inclusion of some comments on surgical complication rates after neoadjuvant immunotherapy for PDAC, if possible.
Reviewer 3 Report
Cancers 2527465
Comments and Suggestions for Authors:
This study was a review for investigating the role of neoadjuvant immunotherapy for localized pancreas cancer by systematic search. The issue authors investigated in this study is very attractive one to be clarified at the present time. Therefore, this review study might be very timely and appropriate review of preliminary studies in this field. This was described well and discussed properly.
1 As the minor point of issue, these preliminary studies that authors cited in this review were small number of studies and also small number of patient’s series. Therefore, nine published trials should be listed as a Table. Authors should prepare the new Table for these 9 published trials in this manuscript. Therein, the number of subjected patients in each study should be also clearly described in a Table.
Cancers 2527465
Comments and Suggestions for Authors:
This study was a review for investigating the role of neoadjuvant immunotherapy for localized pancreas cancer by systematic search. The issue authors investigated in this study is very attractive one to be clarified at the present time. Therefore, this review study might be very timely and appropriate review of preliminary studies in this field. This was described well and discussed properly.
1 As the minor point of issue, these preliminary studies that authors cited in this review were small number of studies and also small number of patient’s series. Therefore, nine published trials should be listed as a Table. Authors should prepare the new Table for these 9 published trials in this manuscript. Therein, the number of subjected patients in each study should be also clearly described in a Table.
